# Free energy landscape of RNA binding dynamics in start codon recognition by eukaryotic ribosomal pre-initiation complex

**Takeru Kameda**[1,2], **Katsura Asano**[3,4,5]*, **Yuichi Togashi**[4,6,7¤]*

**1** Graduate School of Science, Hiroshima University, Higashi-Hiroshima, Hiroshima, Japan, **2** RIKEN Center for Biosystems Dynamics Research (BDR), Wako, Saitama, Japan, **3** Molecular Cellular and Developmental Biology Program, Division of Biology, Kansas State University, Manhattan, Kansas, United States of America, **4** Graduate School of Integrated Sciences for Life, Hiroshima University, Higashi-Hiroshima, Hiroshima, Japan, **5** Hiroshima Research Center for Healthy Aging (HiHA), Hiroshima University, Higashi-Hiroshima, Hiroshima, Japan, **6** Research Center for the Mathematics on Chromatin Live Dynamics (RcMcD), Hiroshima University, Higashi-Hiroshima, Hiroshima, Japan, **7** RIKEN Center for Biosystems Dynamics Research (BDR), Higashi-Hiroshima, Hiroshima, Japan

¤Current address: College of Life Sciences, Ritsumeikan University, Kusatsu, Shiga, Japan
* kasano@ksu.edu (KA); yuichi.togashi@riken.jp (YT)

**Data Availability Statement:** The datasets generated for this study are available from the Zenodo repository doi: 10.5281/zenodo.4577431.

## Abstract

Specific interaction between the start codon, 5'-AUG-3', and the anticodon, 5'-CAU-3', ensures accurate initiation of translation. Recent studies show that several near-cognate start codons (e.g. GUG and CUG) can play a role in initiating translation in eukaryotes. However, the mechanism allowing initiation through mismatched base-pairs at the ribosomal decoding site is still unclear at an atomic level. In this work, we propose an extended simulation-based method to evaluate free energy profiles, through computing the distance between each base-pair of the triplet interactions involved in recognition of start codons in eukaryotic translation pre-initiation complex. Our method provides not only the free energy penalty for mismatched start codons relative to the AUG start codon, but also the preferred pathways of transitions between bound and unbound states, which has not been described by previous studies. To verify the method, the binding dynamics of cognate (AUG) and near-cognate start codons (CUG and GUG) were simulated. Evaluated free energy profiles agree with experimentally observed changes in initiation frequencies from respective codons. This work proposes for the first time how a G:U mismatch at the first position of codon (GUG)-anticodon base-pairs destabilizes the accommodation in the initiating eukaryotic ribosome and how initiation at a CUG codon is nearly as strong as, or sometimes stronger than, that at a GUG codon. Our method is expected to be applied to study the affinity changes for various mismatched base-pairs.

## Author summary

Ribosomes synthesize proteins according to the sequence of nucleotides (A, U, G, and C) in mRNA, translating three nucleotides (codon) into an amino acid. If the reading frame

**Funding:** YT was supported by JSPS KAKENHI Grant Number JP18KK0388, and by JSPS and NRF under the Japan-Korea Basic Scientific Cooperation Program (https://www.jsps.go.jp/, https://www.nrf.re.kr/). KA was supported by K-INBRE program Pilot Grant (https://www.k-inbre.org/), National Institutes of Health [P20 GM103418]; National Institutes of Health R15 grant [GM125671] (https://www.nih.gov/); National Science Foundation Research Grant [1412250] (https://www.nsf.gov/); and JSPS KAKENHI [JP18K19963] (https://www.jsps.go.jp/). The funders had no role in study design, data collection and analysis, decision to publish, or preparation of the manuscript.

**Competing interests:** The authors have declared that no competing interests exist.

is shifted, the resulting amino-acid sequence will be totally different. Hence, the translation should start at an exactly determined position of mRNA. This position is usually indicated by "AUG" (start codon), which is recognized by "CAU" anticodon in tRNA and then translated into methionine. However, translation sometimes initiates at another codon such as "GUG" or "CUG", and its frequency varies depending on the codon. Then, what regulates the possibility of translation-initiation at such a mismatched codon? To answer the question, we employed computer simulation for the structural changes of RNAs and proteins involved in the process. Through the numerical analysis, we estimated how strongly the mRNA (codon) and the tRNA (anticodon) bind to each other and also inferred how they approach. The binding-strength correlates with the initiation frequency observed in experiments, and the approaching pathway could explain the difference. Our result shows the underlying mechanism for the fidelity of translation-initiation, and our method will be applied to the prediction and design of RNA-RNA interactions.

## Introduction

The translation reaction, or mRNA-dependent protein synthesis, is catalyzed by the ribosome, the macromolecular ribonucleoprotein complex [1, 2]. During eukaryotic initiation, the ribosome dissociates into the large (60S) and small (40S) subunits, and the latter binds the methionyl initiator tRNA ($\mathrm{Met} - \mathrm{tRNA}_i^{\mathrm{Met}}$) and mRNA with the help of eukaryotic initiation factors (eIFs) [3, 4]. $\mathrm{Met} - \mathrm{tRNA}_i^{\mathrm{Met}}$ is recruited by eIF2, a heterotrimeric factor that binds the tRNA in a manner dependent on GTP binding. The resulting ternary complex (TC) binds the 40S subunit in the context of multifactor complex (MFC) with eIFs 1, 3 and 5, forming the 43S pre-initiation complex (PIC) [5]. The mRNA is bound by eIF4F through its 5' cap, which then is recruited to the 40S subunit through eIF3 in mammals and eIF5 in yeast [6]. The 48S pre-initiation complex (PIC) thus formed scans for the start codon in the process called scanning. The PIC is primed for scanning by the eIF5-catalyzed GTP hydrolysis for eIF2; the products, GDP and Pi, stay bound to eIF2 during scanning. The start codon base-pairing with the $\mathrm{Met} - \mathrm{tRNA}_i^{\mathrm{Met}}$ anticodon in the P-site allows the 40S subunit to stall at the start codon and then join the 60S subunit after most bound eIFs are released in conjunction with Pi release from eIF2 [7, 8]. The resulting 80S initiation complex accepts an amino-acyl tRNA in the A-site to begin the translation elongation cycle.

The fidelity of start codon recognition is regulated by eIF1A and eIF1 that bind the 40S subunit A-site and P-site, respectively [9, 10]. Essentially, these factors regulate the conformational changes of the PIC. Thus, the N-terminal tail of eIF1A interacts with the codon-anticodon base-pairs to stabilize its closed conformation for initiation [11]. By contrast, eIF1 stabilizes its open, scanning-competent conformation by physically impeding the P-site accommodation of the mismatched codon-anticodon base-pairs [5]. Upon start codon selection by the PIC, eIF1 is released to permanently stabilize the closed state [12, 13]. The PIC-bound eIFs directly interact with eIF1 to control the balance of its binding and release, thereby keeping the level of initiation accuracy appropriate for eukaryotic cell function [6, 14–16]; for review, see [17].

Despite the aforementioned mechanisms to ensure the accurate initiation of translation, most eukaryotes tested allow translation initiation from near-cognate start codons (start codons with one-base substitution in the AUG codon), for example, CUG and GUG, at a low frequency [18–21]. However, not all the near-cognate start codons can initiate translation at an equal rate. Among the most puzzling is the observation that GUG serves as a poor initiation site, even though the G residue in the 1st position can potentially wobble-base-pair with the U

residue in the anticodon. In agreement with a wobble base-pairing, GUG serves as a normal initiation site in prokaryotes (Bacteria and Archaea), which possess fewer initiation factors than in eukaryotes [17]. Moreover, CUG is considered as the strongest near-cognate start codon in eukaryotes, even though it is not a start codon in prokaryotes [17]. To solve these conundrums, it is crucial to probe the stability of codon-anticodon interactions in the P-site. As it is difficult to experimentally measure these interactions at an atomic resolution, computational analysis offers an effective solution [22, 23]. Specifically, molecular interactions are represented by the free energy profiles, which in this case depend on the nucleotide constitutions of codon and anticodon, and thereby determine the frequency of the bound state. Thus, computational estimation of their free energy profiles provides an insight into the mechanism of translation initiation by various start codons.

Previous molecular dynamics (MD) simulation studies estimated energy gap between AUG and mismatched codons by computing free energy scores of interaction between these codons by free energy perturbation (FEP) [24]. The work identified CUG as the strongest near-cognate start codon. Moreover, it demonstrated eIF1's contribution in discriminating from near-cognate start codons. However, there are mainly two caveats in the FEP approach. First, except for the conformation of AUG found within the PIC structure, the geometry of near-cognate start codon was conjectured merely through substituting a base with the corresponding base in the AUG. Second, information on the binding process was missing. As single-strands of RNA are flexible [25], their structural changes may be diverse. Although the binding free energy is independent of the transition path and thus could be calculated by the FEP method, it is crucial to estimate structural changes involved in the binding process in order to understand the selective binding mechanism.

In this study, we employed adaptive biasing force (ABF) method [26–28] in order to overcome these problems. This method explores codon-anticodon binding free energy by using systematic reaction coordinates. By using distances of each triplet base-pair as the reaction coordinates, we generated the free energy profiles of base-pairing interactions between $\text{Met} - \text{tRNA}_i^{\text{Met}}$ anticodon and cognate (AUG) or near-cognate (CUG or GUG) start codons. Our results provide structural insights related to a strong penalty for placing GUG codon in the P-site and a permissiveness of CUG as a potential start codon in eukaryotes.

## Materials and methods

### Simulation procedure

We referred to previous study of start codon recognition in eukaryotic translation initiation using all-atom molecular dynamics (MD) simulations [24]. To reconstruct codon-anticodon interaction in solution, we employed open pre-initiation complex (PIC) structure (PDB ID: 3J81) determined by CryoEM [11]. To reduce computational cost for MD simulation, we extract atoms within 25 Å from N1 atom in the middle base of anticodon in tRNA molecule [24]. Then, nucleotides were edited to reconstruct PIC models involving target codons in our study (Table 1). When editing the nucleotide (e.g. AUG → GUG), first, atoms except N1 and

**Table 1. Codons modeled in our simulation.** Nucleotide sequence and reference atoms to define the codon-anticodon distance.

| Codon | 1st Nucleotide | | 2nd Nucleotide | | 3rd Nucleotide | |
|---|---|---|---|---|---|---|
| AUG | A | N1 and N6 | U | N3 and O4 | G | N1, N2, and O6 |
| GUG | G | N1, N2, and O6 | U | N3 and O4 | G | N1, N2, and O6 |
| CUG | C | N3, N4, and O2 | U | N3 and O4 | G | N1, N2, and O6 |

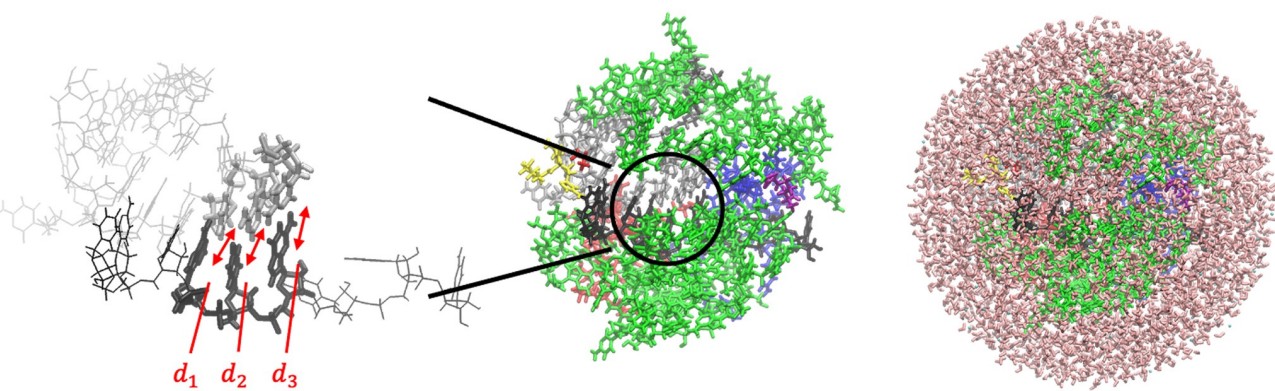

**Fig 1. PIC structure models and reaction coordinates in this study.** (Left) mRNA and tRNA around the codon-anticodon base-pairs (Table 1) are shown in black and silver, respectively. Distances $d_1$, $d_2$ and $d_3$ used as reaction coordinates are indicated. (Center) rRNA segments (split) are shown in green, and protein components are shown in other colors. (Right) Water molecules are shown in pink, which enclose the system and form a sphere.

N9 in base group, sugar group, and phosphate group were deleted. Then, coordinates of missing atoms were inferred. All histidine residues were configured as $\epsilon$-protonated. These molecules were soaked into a 36 Å radius water sphere (Fig 1), neutralized by $K^+$, and added 150 mM KCl. TIP3P water model was employed. VMD [29] was used to infer missing atom coordinates, solvate the model, and visualize the structure throughout the study.

All simulations were carried out using NAMD (version 2.13 multi-core) [30]. CHARMM36 force-field (July 2019 update) was used [31, 32]. Multilevel summation method (MSM) electrostatics [33] was employed. Cutoff at 12 Å (with switching from 10 Å) was applied to non-bonded interactions. Temperature and pressure were set at 310 K and 1 atm, respectively; Langevin thermostat (damping coefficient: 5/ps) and Langevin-piston barostat were adopted. All C1' (nucleotide) and $C^\alpha$ (amino acid) atoms farther than 22 Å from the center of the system (i.e. water sphere) were restrained at their initial positions, and water molecules crossing the boundary of water sphere (radius 36 Å) were restrained. Harmonic potential functions with spring constant 10 pN/Å were adopted as the restraint of molecules. After energy minimization (10,000 steps), the system was equilibrated for 10 ns, and then simulated for 1 $\mu$s (timestep: 2 fs); the biasing force was applied only after collecting 200 samples in the bin. Each model (Table 1) was simulated five times.

## Multi dimensional adaptive biasing force (ABF) method

We performed adaptive biasing force (ABF) molecular dynamics method [26–28] to evaluate multi-dimensional free energy profiles in terms of $d_1$, $d_2$, and $d_3$, which were defined as the distances of the 1st, 2nd, and 3rd base-pairs in Å, respectively (Table 1). Specifically, these $d_i$ were evaluated as distance between the centers of hydrogen donor and acceptor atoms of codon and anticodon (Table 1 shows the atoms in a mRNA segment). Each $d_i$ was sampled over $4.0 \leq d_i \leq 9.0$ with bin width $\Delta d$ = 0.5 Å, and was restrained by harmonic bonds with spring constant 10 pN/Å if $d_i$ crosses lower boundary (3.0 Å) or upper boundary (10.0 Å).

The Gibbs free energy profile $G(d_1, d_2, d_3)$ with respect to three variables $d_1$ to $d_3$ was obtained through the analysis of ABF results. The probability $P(d_1, d_2, d_3)$ of each state

obeys Eq 1:

$$P(d_1, d_2, d_3) := \frac{\exp\left(-\dfrac{G(d_1, d_2, d_3)}{k_B T}\right)}{\sum_{4.0 \leq d_1, d_2, d_3 \leq 9.0} \exp\left(-\dfrac{G(d_1, d_2, d_3)}{k_B T}\right)}. \tag{1}$$

Furthermore, $P(d_1, d_2, d_3)$ was averaged over the simulation trials for each model, and then $G(d_1, d_2, d_3)$ was evaluated as Eq 2:

$$G(d_1, d_2, d_3) = -k_B T \ln P(d_1, d_2, d_3) + const., \tag{2}$$

assuming $\min\{G(d_1, d_2, d_3) | 4.0 \leq d_1, d_2, d_3 \leq 9.0\} = 0$.

Then, to evaluate the free energy difference between the codon-anticodon bound and unbound states, we defined the free energy scores $G_{\text{bound}}$ and $G_{\text{unbound}}$, and their gap $\Delta G_{\text{binding}}$ as Eq 3:

$$G_{\text{bound}} := \frac{\sum_{4.0 \leq d_1, d_2, d_3 \leq 6.0} G(d_1, d_2, d_3) P(d_1, d_2, d_3)}{\sum_{4.0 \leq d_1, d_2, d_3 \leq 6.0} P(d_1, d_2, d_3)}$$

$$G_{\text{unbound}} := \frac{\sum_{7.0 \leq d_1, d_2, d_3 \leq 9.0} G(d_1, d_2, d_3) P(d_1, d_2, d_3)}{\sum_{7.0 \leq d_1, d_2, d_3 \leq 9.0} P(d_1, d_2, d_3)} \tag{3}$$

$$\Delta G_{\text{binding}} := G_{\text{bound}} - G_{\text{unbound}}$$

Here we defined the bound and unbound states as $\forall i : 4.0 \leq d_i \leq 6.0$ and $\forall i : 7.0 \leq d_i \leq 9.0$, respectively (Fig 2). The distance range for the bound state corresponds to the codon-anticodon (AUG-CAU) structure [11], and that for the unbound state is based on a previous research that described unbound conformation of the complex [34]. $G_{\text{bound}}$ and $G_{\text{unbound}}$ were hence weighted average of $G(d_1, d_2, d_3)$ in the ranges of bound and unbound states, respectively (Eq 3). Note that $G_{\text{bound}}$, $G_{\text{unbound}}$, and $\Delta G_{\text{binding}}$ were obtained from $G(d_1, d_2, d_3)$ either in each simulation trial or over five trials for the same model.

To visualize the profiles, $G(d_1, d_2, d_3)$ was projected (marginalized) onto 2-dimension, as $G(d_1, d_2)$ in Eq 4. Projected profile $P(d_1, d_2)$ was obtained from $G(d_1, d_2)$.

$$G(d_1, d_2) := \frac{\sum_{4.0 \leq d_3 \leq 9.0} G(d_1, d_2, d_3) P(d_1, d_2, d_3)}{\sum_{4.0 \leq d_3 \leq 9.0} P(d_1, d_2, d_3)}$$

$$P(d_1, d_2) := \frac{\exp\left(-\dfrac{G(d_1, d_2)}{k_B T}\right)}{\sum_{4.0 \leq d_1, d_2 \leq 9.0} \exp\left(-\dfrac{G(d_1, d_2)}{k_B T}\right)} \tag{4}$$

$G(d_1, d_3)$, $G(d_2, d_3)$ and $P(d_1, d_3)$, $P(d_2, d_3)$ were defined in the same way.

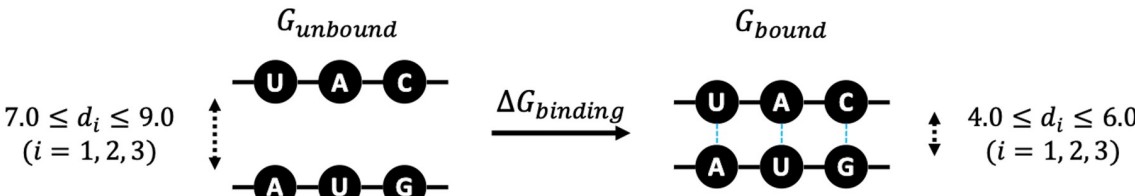

**Fig 2. Binding free energy.** Schematic representation of $G_{\text{bound}}$, $G_{\text{unbound}}$, and $\Delta G_{\text{binding}}$ (Eq 3).

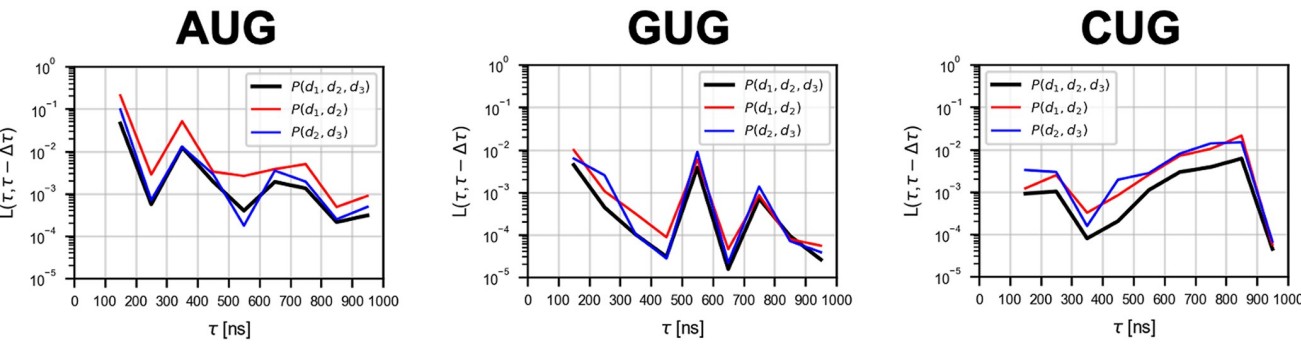

**Fig 3. Convergence of free energy profiles with time evolution.** The $L(\tau, \tau - \Delta\tau)$ of $P(d_1, d_2, d_3)$, $P(d_1, d_2)$, and $P(d_2, d_3)$ are shown (see Eq 5; $\Delta\tau$ = 100 ns). The data points are plotted between the two consecutive time points $(\tau - \Delta\tau)$ and $\tau$.

### Convergence of free energy profiles

Probability $P(d_1, d_2, d_3)$ was obtained at equal intervals $\Delta\tau$ (= 100 ns) through the ABF simulation. We evaluated the convergence of $P(d_1, d_2, d_3)$ with increasing simulation time steps, as follows. To compare profiles of $P(d_1, d_2, d_3)$ at two different time points $(\tau_1, \tau_2)$, we employed squared-error function of two probability distributions. These profiles of different time points $\tau_1$ and $\tau_2$ were represented by $P(d_1, d_2, d_3; \tau_1)$ and $P(d_1, d_2, d_3; \tau_2)$, respectively. Then, the squared-error function $L(\tau_1, \tau_2)$ was calculated as Eq 5:

$$L(\tau_1, \tau_2) := \sum_{4.0 \leq d_1, d_2, d_3 \leq 9.0} \left( P(d_1, d_2, d_3; \tau_1) - P(d_1, d_2, d_3; \tau_2) \right)^2 \tag{5}$$

The convergence of $P(d_1, d_2, d_3)$ at $\tau$ [ns] was tested by $L(\tau, \tau - \Delta\tau)$ (Fig 3). The convergence of $P(d_1, d_2)$, $P(d_2, d_3)$, and $P(d_1, d_3)$ was evaluated in the same way.

### Reconstruction of averaged structure

Typical structures, or atomic coordinates corresponding to a specific reaction coordinate $(d_1, d_2, d_3)$, were obtained by averaging the sampled atomic coordinates as follows. Here, we assumed that the reaction coordinate $(d_1, d_2, d_3)$ is represented by $(\tilde{d}_1, \tilde{d}_2, \tilde{d}_3)$ if Eq 6 is satisfied.

$$\tilde{d}_i - \frac{1}{2}\Delta d \leq d_i < \tilde{d}_i + \frac{1}{2}\Delta d \ (i = 1, 2, 3) \tag{6}$$

Then, each atomic coordinate was averaged over all the snapshots (sampled at 10 ps intervals) corresponding to the representative reaction coordinate $(\tilde{d}_1, \tilde{d}_2, \tilde{d}_3)$.

## Results

### Binding free energy

We evaluated the free energy scores of bound and unbound states ($G_{\text{bound}}$ and $G_{\text{unbound}}$; see Eq 3) averaged over five simulation trials (S1 Fig), and presented their difference $\Delta G_{\text{binding}}$ in Fig 4. In the case of the cognate start codon AUG, $\Delta G_{\text{binding}}$ must be negative to stabilize the initiation of translation, and was indeed $\sim -4\,k_B T$. In contrast, the GUG codon, less frequently used as a start codon [17, 20, 21], showed a positive $\Delta G_{\text{binding}} \sim 2\,k_B T$. For the CUG codon, which is considered as a stronger start codon than GUG [17, 20, 21], $\Delta G_{\text{binding}}$ showed

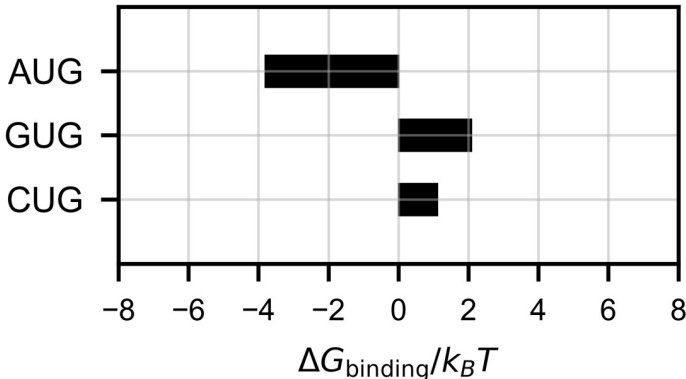

**Fig 4. Estimated binding free energy.** $\Delta G_{\text{binding}}$ scores (Eq 3; see also Fig 2) are shown. The scores were obtained from $P(d_1, d_2, d_3)$ averaged over five simulation trials for each model.

an intermediate value $\sim 1 \, k_B \, T$. Thus, $\Delta G_{\text{binding}}$ accounts for observed initiation rates from the respective start codons. Note that $\Delta G_{\text{binding}}$ could alternatively be obtained from individual simulation trajectories (S2 Fig); assuming each trial as a sample, we confirmed the significance of $\Delta G_{\text{binding}}$ between AUG and GUG by Welch's $t$-test ($p = 0.044$).

## Free energy landscape for the binding process

To analyze the base-pair binding dynamics in detail, we constructed projected profiles $G(d_1, d_2)$, $G(d_1, d_3)$, and $G(d_2, d_3)$ from $G(d_1, d_2, d_3)$ (see Materials and methods), as shown in Fig 5. We inferred the transition dynamics from the free energy profiles; Fig 6 shows the suggested paths and their schematics.

In the case of the AUG start codon, the following transitions were expected in the AUG-CAU dynamics in equilibrium (Fig 6, left). Starting from the bound state, $d_3$ shows large fluctuations while $d_1$ and $d_2$ show small ones ($R_3^{\text{AUG}}$) (Fig 5B and 5C). $d_1$ and $d_2$ are bistable (bound and unbound), and once the 3rd G:C base-pair is broken (large $d_3$), the 2nd U:A base-pair may become unbound ($R_2^{\text{AUG}}$ to the large $d_2$ state) (Fig 5B). Only after that, the 1st A:U base-pair dissociates ($R_1^{\text{AUG}}$ to the large $d_1$ state) (Fig 5A); this is expected to occur less frequently due to the higher barrier than those for $R_2^{\text{AUG}}$ and $R_3^{\text{AUG}}$. Starting from the unbound state and reversing the process above, the AUG codon should bind to the CAU anticodon from the side of the 1st A:U base-pair, followed by the 2nd and then 3rd base-pairs (Fig 6, left). This result suggests that the recognition of the 1st A:U base-pair is very important for the accurate start codon recognition, in agreement with the role and location of eIF1 in the P-site [11, 12] (see below).

In the case of the GUG codon (Fig 6, right), $d_1$ and $d_2$ show the transition ($R_1^{\text{GUG}}$) between two distinct (metastable) states (both bound and both almost unbound) (Fig 5D), while $d_3$ is mostly high (Fig 5E and 5F). Binding of the 3rd base-pair (transition to lower $d_3$) is possible but less frequent, and simultaneous binding of the 1st and 3rd base-pairs is rare (Fig 5F). As expected, the affinity of the 1st base-pair (wobble G:U) is lower than the case of AUG. This result is consistent with infrequent GUG initiation observed in the previous works [17, 20, 21].

In the case of the CUG codon, many meta-stable states were observed as shown in Fig 5G–5I. Transition paths seem to be more complicated than the AUG and GUG cases. Although concurrent binding of the 2nd and 3rd base-pairs (lower $d_2$ and $d_3$) is possible (Fig 5H), the 1st base-pair cannot form simultaneously with these other base-pairs (Fig 5G and 5I), which

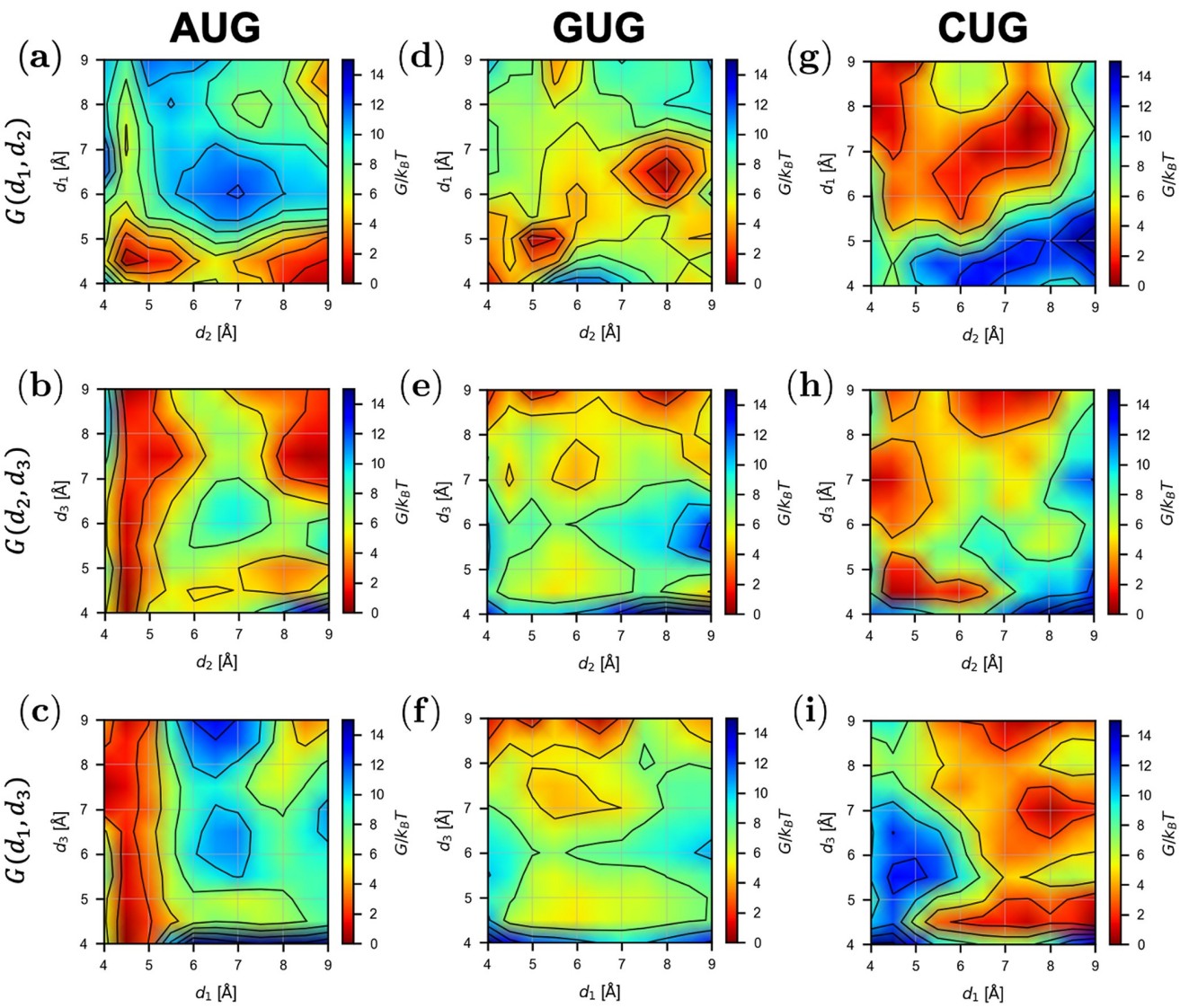

**Fig 5. Projected free energy profiles.** Profiles of $G(d_1, d_2)$, $G(d_2, d_3)$, and $G(d_1, d_3)$ obtained from Eq 4 for each model is shown by contour plots.

makes the CUG pairing unstable compared to AUG base-pairing. Overall, however, the binding free energy $\Delta G_{\text{binding}}$ is lower for CUG than for GUG (Fig 4) (see below). Note that, technically, the rugged free energy landscape (Fig 5G–5I) demanded more computational cost for the ABF sampling, as suggested by the slow convergence shown in Fig 3.

## Binding dynamics from the structural views

To visualize the bound structures and consider the mechanism underlying the abovementioned results (Figs 5 and 6), we evaluated the averaged structure of the bound state for each model (see Eq 6 in Materials and Methods). The averaged structures and schematics of the codon and anticodon are shown in Fig 7.

In the case of AUG, the averaged bound-state structure is ordered and tightly bound. It is reasonable, as it is the correct start codon, and the binding free energy is negative (Fig 4). Note

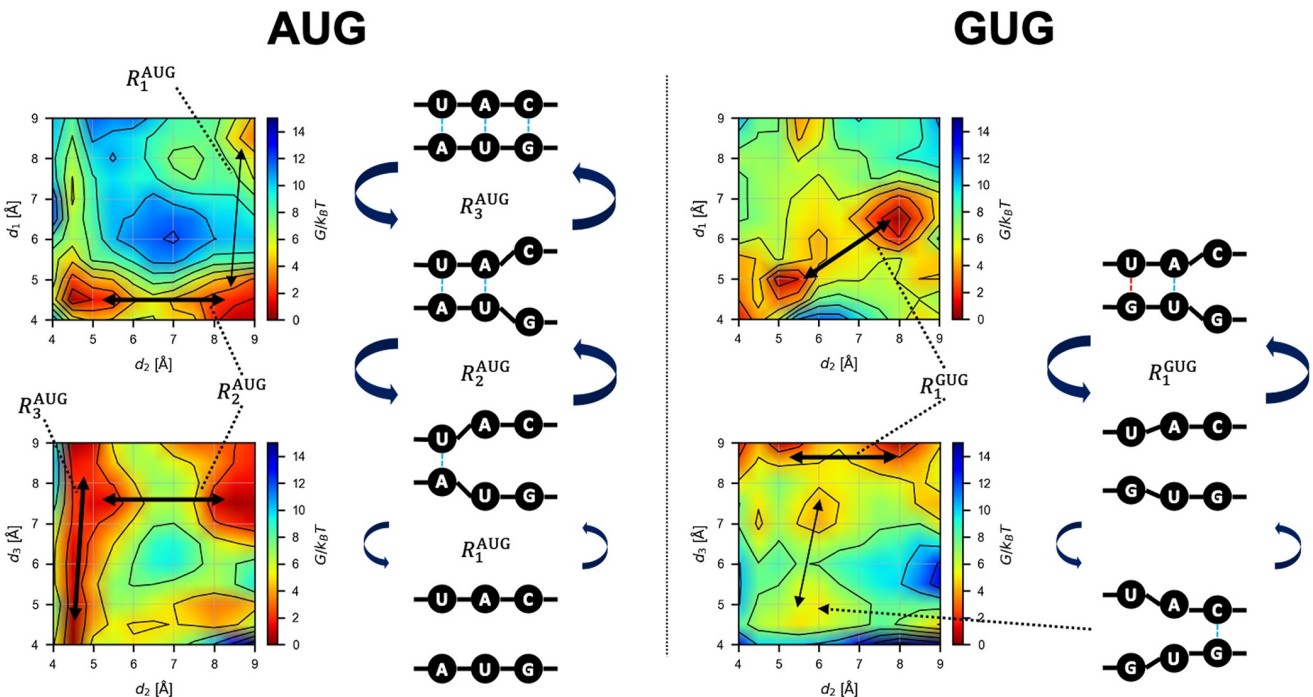

**Fig 6. Schematics of the base-pair binding dynamics.** Conformational changes inferred from the free energy landscape (Fig 5). The transition path $R_n^\bullet$ ($\bullet$ is AUG or GUG) is shown by black arrows (thick arrow: frequent transition; thin arrow: less frequent transition). Blue and red dotted lines indicate Watson-Crick or wobble base-pairing, respectively. In left, AUG pairing is described as being stabilized by the first position. Subsequently, the binding propagates into the second and the third position. In right, GUG pairing is described as bistable states; the base-pairing between GU and the corresponding bases in the anticodon, and the base pairing between the 3rd G and the C residue of the anticodon. Thus, a complete base-triplet interaction is unstable.

that eIF1 and eIF1A molecules (shown in red and blue in Fig 7, respectively) are present near the AUG-CAU base-pair. It was experimentally suggested that these proteins contribute to the accurate start codon recognition [5, 9–13] (see Fig 8).

By contrast, the structure of the GUG-CAU base-pair is disordered (Fig 7). The mismatched bases (the 1st G:U) avoid each other (rather than forming a wobble base-pair) and the uracil in tRNA tilts toward the 2nd U:A base-pair. The directions of the 2nd and 3rd base-pairs were consequently affected, resulting in the unstable bound state (Fig 4). Although the projected free energy profile $G(d_1, d_2)$ (Fig 5D) suggests cooperative binding of the 1st and 2nd base-pairs (Fig 7), the 3rd base-pair is mostly separate, which may prevent the recognition of the GUG start codon.

In the case of CUG, the structure is relatively ordered (Fig 7). Although the 1st C:U base-pair is mismatched, cytosine is smaller than guanine and adenine (purine bases), which may mitigate steric hindrance at the 1st position. As shown in Fig 5G to 5I, many meta-stable conformations are possible, which we propose to be attributed to combinations of bound and unbound conformations of the base-pairs. It is therefore reasoned that some near-bound states can occasionally allow translation initiation at this codon (Fig 7, right).

## Discussion

The binding free energy $\Delta G_\text{binding}$ evaluated by our method was qualitatively consistent with experimental observation, as described in Fig 4. We compared the AUG start codon and two

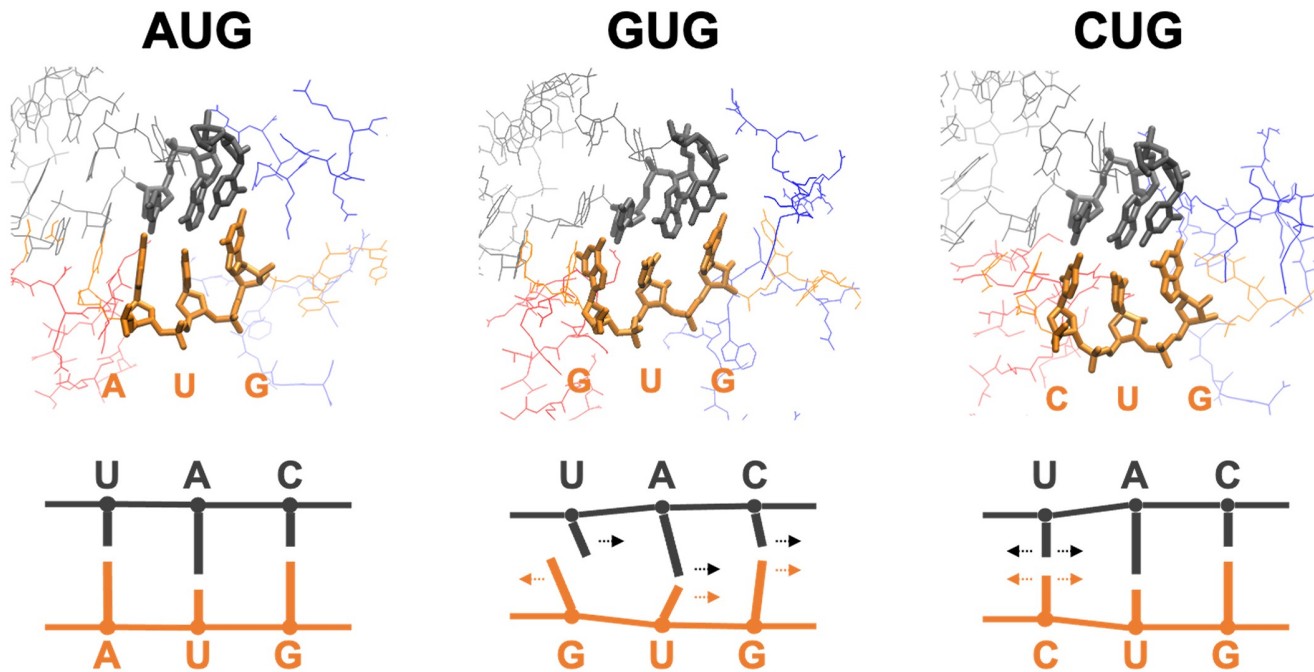

**Fig 7. Typical bound structures.** Averaged structures corresponding to reaction coordinate $(\tilde{d}_1, \tilde{d}_2, \tilde{d}_3) = (4.5, 4.5, 4.5)$ (see Eq 6). In the structures on top, nucleotides of the codon (orange) and anticodon (gray) are drawn by thick lines. Red and blue lines are parts of eIF1 and eIF1A, respectively. Note that the structures of GUG and CUG pairing (middle and right) are much less stable than that of AUG pairing (left) (see Fig 4). The schematics on the bottom describe the direction of each base relative to the paired base.

near-cognate start codons (Table 1). Assuming that $G_{unbound}$ is common for all the models, i.e. the free energy of the unbound state is independent of the codon, $G_{bound}$ and $\Delta G_{binding}$ are equivalent. The difference, or penalty, of binding free energy ($\Delta\Delta G$) induced by AUG → GUG and AUG → CUG substitution was $\simeq 6 \, k_B T$ (3.6 kcal/mol) and $\simeq 5 \, k_B T$ (3.0 kcal/mol), respectively. This result is largely consistent with another computational approach using the free energy perturbation (FEP) [24].

In contrast to the previous work, however, our ABF-based approach provided not merely the binding free energy but information on the nucleic acid binding dynamics represented by the free energy landscape (Figs 5 and 6). The free energy profiles shown in Fig 5A–5C suggested an unexpected stability of the 1st A:U base-pair, compared to the 3rd G:C base-pair. According to the free energy profile of AUG binding, dissociation of the triplet base-pairs starts at the 3rd G:C (Fig 6, $R_3^{AUG}$). In the open PIC model that is suggested to occur during the scanning process prior to start codon recognition [11], the tRNA is not perpendicularly attached to the mRNA, in contrast to the P-site tRNA positioning during the elongation phase. This conformation appears to allow the 5'-side (i.e. cytosine side) of the anticodon to curve away from the start codon, suggesting a stretching force towards the tRNA side (Fig 8A). We propose that this stretching decreases the affinity of the 3rd G:C base-pair during the scanning process (Fig 8A). In contrast, the affinity of the 1st A:U base-pair is likely to be increased by interaction with eIF1, so is that of the 2nd U:A base-pair by eIF1A, as proposed previously [5, 9–13] (Fig 8A). In strong agreement with the role of eIF1 in stabilizing the 1st A:U base-pair, our averaged simulation structure indeed positions Asn-34 and Arg-36 in its proximity (Fig 8B). In fact, the residues Asn-34:Gly-35:Arg-36, termed $\beta$-hairpin loop 1, is absolutely

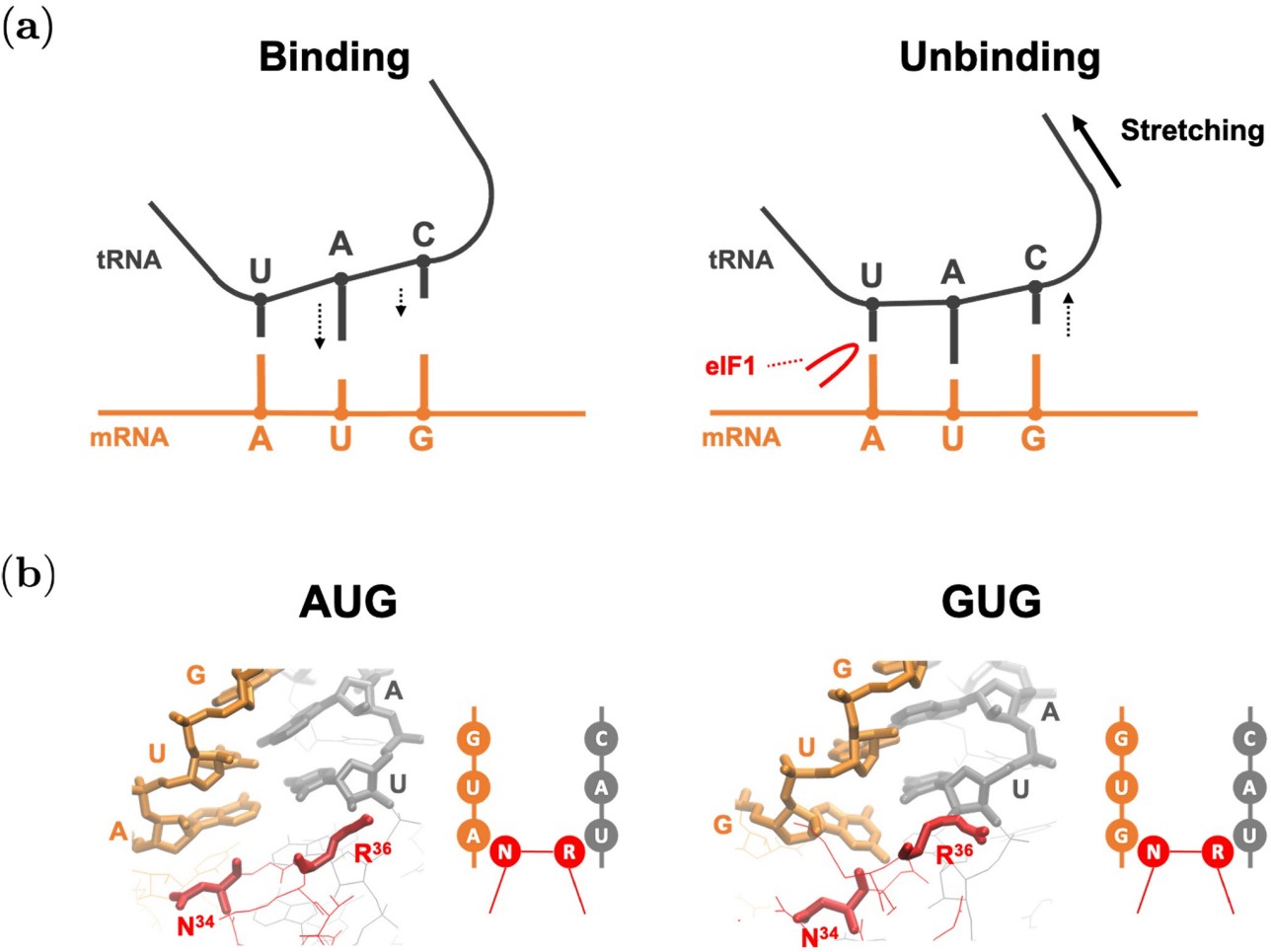

**Fig 8. Binding mechanism conjectured by this study.** mRNA and tRNA are drawn by orange and gray, respectively. Red lines show parts of eIF1. (a) Schematics of binding and unbinding dynamics of AUG pairing in the P-site. (Left) AUG binding is facilitated by the stabilization of the first base-pair, followed by base-pairing of the 2nd and 3rd positions (downward arrows). (Right) AUG unbinding appears to be facilitated by dissociation of the 3rd position (upward arrow) through stretching (thick arrow) imposed by the open ($P_{OUT}$) conformation of the PIC [11]. Here, eIF1 $\beta$-hairpin loop (red) is proposed to prevent dissociation of the base-pairing at the 1st position [35]. (b) Averaged structures corresponding to reaction coordinate $(\tilde{d}_1, \tilde{d}_2, \tilde{d}_3) = (4.5, 4.5, 4.5)$ (see Eq 6), and their schematic representations. Nucleotides of the codon and anticodon, and $N^{34}$ (Asn-34) and $R^{36}$ (Arg-36) are drawn by thick lines.

conserved from yeast to human. Mutations altering Asn-34 and Arg-36 display significant increase in UUG initiation [35], in agreement with their crucial role in maintaining open scanning-competent PIC conformation.

The free energy landscape of GUG-anticodon base-pairs (Fig 5D–5F) and its averaged simulation structure in the P-site (Figs 7 and 8B) also suggest that the same structural restriction in turn prevents G:U pairing at the 1st position, that otherwise occurs frequently in its free form. The disordered 3rd G:C base-pair seen with the GUG structure appears to be consistent with this idea (Fig 7). Since we did not observe a strong disorder in CUG-anticodon structure (Fig 7), we propose that the near-cognate start codon usage characteristic of eukaryotic initiation is mostly explained by a strong perturbation on GUG accommodation in the P-site due to steric restriction imposed by eIF1 $\beta$-hairpin loop. In agreement with this thesis, the level of

CUG initiation is just equivalent to that of GUG initiation in yeast *S. cerevisiae* [19] (and our personal observations), although the former is significantly stronger than the latter in various distinct contexts in human cells [21].

The adaptive biasing force (ABF) method adopted in this study was previously used to evaluate free energy profiles of molecular dynamics in other biomolecular systems [26–28]. Among many extended ensemble simulation methods, we chose ABF to mitigate the difficulties in studying nucleic acid (DNA and RNA) interactions. An example of such difficulty is encountered when high temperature is applied in replica exchange molecular dynamics [36] that causes complete separation of nucleic acid strands, which cannot revert to the original structure within the simulation timescale. To overcome this problem, extended simulation methods without destroying the structure should be sought, and the ABF can offer a solution. We hope that further development of this approach can also contribute to the improvement of analysis and prediction of RNA structural dynamics [37, 38].

## Conclusion

In this study, we proposed a computational method to obtain multi-dimensional free energy profiles for codon-anticodon base-pairing by adaptive biasing force (ABF) molecular dynamics [26–28]. This reaction-coordinate-based analysis method provided the equilibrium profiles of base-pair binding dynamics depending on ribonucleotide sequence—start codon-anticodon base-pairing in this case. Our method successfully detected the changes in the free energy landscape (Figs 5 and 6) induced by site-specific nucleotide substitution in the start codon (e.g. AUG → GUG) and offered a mechanistic explanation for how such changes led to perturbation in initiation frequency from the altered start codons.

## Supporting information

**S1 Fig. Estimated free energy of bound and unbound states.** Red and blue bars correspond to $G_{bound}$ and $G_{unbound}$ (Eq 3), respectively (see the schematics in Fig 2). The scores were obtained from $P(d_1, d_2, d_3)$ averaged over five simulation trials for each model.
(TIF)

**S2 Fig. Binding free energy for individual simulation trials.** Error bars show the mean ± S.E. M. of $\Delta G_{binding}$ obtained from each simulation trajectory. Note that the mean $\Delta G_{binding}$ here is different from the $\Delta G_{binding}$ score in Fig 4, for which $P(d_1, d_2, d_3)$, not $G$, was averaged over simulation trials; this SI figure is presented only for reference and Fig 4 is physically relevant.
(TIF)

## Acknowledgments

The authors are grateful to A. Awazu and C. Singh for fruitful discussions. The computation was carried out using the computer resource offered under the category of General Projects by Research Institute for Information Technology, Kyushu University.

## Author Contributions

**Conceptualization:** Takeru Kameda, Katsura Asano, Yuichi Togashi.

**Formal analysis:** Takeru Kameda, Katsura Asano, Yuichi Togashi.

**Funding acquisition:** Katsura Asano, Yuichi Togashi.

**Investigation:** Takeru Kameda.

**Methodology:** Takeru Kameda, Yuichi Togashi.

**Project administration:** Katsura Asano, Yuichi Togashi.

**Resources:** Yuichi Togashi.

**Software:** Takeru Kameda.

**Supervision:** Katsura Asano, Yuichi Togashi.

**Visualization:** Takeru Kameda.

**Writing – original draft:** Takeru Kameda, Katsura Asano, Yuichi Togashi.

**Writing – review & editing:** Takeru Kameda, Katsura Asano, Yuichi Togashi.

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
