## [Decision Letter · Decision Letter 0]

24 Apr 2021

Dear Dr. Togashi,

Thank you very much for submitting your manuscript "Free Energy Landscape of RNA Binding Dynamics in Start Codon Recognition by Eukaryotic Ribosomal Pre-Initiation Complex" for consideration at PLOS Computational Biology. As with all papers reviewed by the journal, your manuscript was reviewed by members of the editorial board and by several independent reviewers. The reviewers appreciated the attention to an important topic. Based on the reviews, we are likely to accept this manuscript for publication, providing that you modify the manuscript according to the review recommendations.

Sincerely,

Eugene I. Shakhnovich

Guest Editor

PLOS Computational Biology

Nir Ben-Tal

Deputy Editor

PLOS Computational Biology

[LINK]

Reviewer's Responses to Questions

**Comments to the Authors:**

Reviewer #1: In this work, Kameda et al. performed adaptive biasing force molecular dynamics simulation to understand to free energy profiles for mRNA (codon) and the tRNA (anticodon) pairing. They further inferred the transition dynamics based on the free energy landscapes and explored atomic-level details of the start codon and near-cognate codon and their corresponding anticodon. Overall, this work enhanced the understanding of the mechanism of start codon and near-cognate codon in initiation of translation.

I only have several comments:

1. In Figure 3, it seems the curves are fluctuating instead of being more smoothed. Is this the nature of the simulation technique? Especially for CUG, it seems the curve keeps going up and drops sharply close to the end. Also is that possible to have error bars on these curves?

2. The authors mentioned the free energy scores of bound and unbound states were averaged from five simulation trials. But I don't see any error/uncertainty estimation in Figure S1 and 4.

3. There are multiple panels in Figure 5. It would be better if they are all labeled (e.g., with letters) so that the authors could be more specific when they refer to these panels in the main text.

Reviewer #2: In this paper, Kameda et al report the free energy landscape of the interaction of the AUG start codon along with two non-cognate start codons (GUG and CUG) with the corresponding anti-codon on the ribosome in the eukaryotic pre-initiation complex using adaptive biasing force (ABF) molecular dynamics simulation. Contrary to earlier methods that have used free energy perturbation, ABF explicitly takes into account the conformations of the base pairs. The authors simulate several combinations of distances between base pairs in the three codon-anticodon pairs to compute their free energy landscape, which allows them to predict preferred transition pathways of between bound and unbound states. The authors demonstrate that between the two non-cognate start codons, GUG is less favorable than CUG, despite presence of the G-U wobble base pair in the former, and in the process are able to recapitulate the experimental observation that CUG is more frequently used than GUG. Additionally, the authors suggest a probable mechanism as to why the commonly observed GU wobble base pairing is less favorable in this case, predominantly due to steric hindrance caused by the beta-hairpin of eIF1. This work is important to mechanistically understand why certain codons are favored at the start position, and the methodology holds promise towards investigation of future RNA structural dynamics.

Comment: Based on the free landscape for GUG (Fig 6), the R1(GUG) structure (shown on top) seems more favored than where only the third base is paired (lower panel). However, in Fig 7, the structure that is shown for GUG has a different structure where the second and third base are paired. Unless I have missed something, this seems confusing.

Minor:

1. In Fig 6, some base pairs are shown by blue dotted lines, while some are in red. It is not clear what this color code represents.

2. Figure legends is most cases would benefit from more details, about what different symbols and arrows suggest, and a small note about the significance of the figure, if possible. This would prevent a lot of back and forth reading.

**Have the authors made all data and (if applicable) computational code underlying the findings in their manuscript fully available?**

Reviewer #1: Yes

Reviewer #2: Yes

PLOS authors have the option to publish the peer review history of their article (what does this mean?). If published, this will include your full peer review and any attached files.

Reviewer #1: No

Reviewer #2: No

Figure Files:

Data Requirements:

Reproducibility:

References:

---

## [Editor Report · Decision Letter 1]

12 May 2021

Dear Dr. Togashi,

We are pleased to inform you that your manuscript 'Free Energy Landscape of RNA Binding Dynamics in Start Codon Recognition by Eukaryotic Ribosomal Pre-Initiation Complex' has been provisionally accepted for publication in PLOS Computational Biology.

Best regards,

Eugene I. Shakhnovich

Guest Editor

PLOS Computational Biology

Nir Ben-Tal

Deputy Editor

PLOS Computational Biology

---

## [Editor Report · Acceptance letter]

10 Jun 2021

PCOMPBIOL-D-21-00485R1 

Free Energy Landscape of RNA Binding Dynamics in Start Codon Recognition by Eukaryotic Ribosomal Pre-Initiation Complex

Dear Dr Togashi,

I am pleased to inform you that your manuscript has been formally accepted for publication in PLOS Computational Biology. Your manuscript is now with our production department and you will be notified of the publication date in due course.

With kind regards,

Katalin Szabo
